# META-LEARNING TRANSFERABLE ACTIVE LEARNING POLICIES BY DEEP REINFORCEMENT LEARNING

## ABSTRACT

Active learning (AL) aims to enable training high performance classifiers with low annotation cost by predicting which subset of unlabelled instances would be most beneficial to label. The importance of AL has motivated extensive research, proposing a wide variety of manually designed AL algorithms with diverse theoretical and intuitive motivations. In contrast to this body of research, we propose to treat active learning algorithm design as a meta-learning problem and learn the best criterion from data. We model an active learning algorithm as a deep neural network that inputs the base learner state and the unlabelled point set and predicts the best point to annotate next. Training this active query policy network with reinforcement learning, produces the best non-myopic policy for a given dataset. The key challenge in achieving a general solution to AL then becomes that of learner generalisation, particularly across heterogeneous datasets. We propose a multi-task dataset-embedding approach that allows dataset-agnostic active learners to be trained. Our evaluation shows that AL algorithms trained in this way can directly generalise across diverse problems.

## 1 INTRODUCTION

In many applications, supervision is costly relative to the volume of data. In these settings active query selection methods can be invaluable to predict which instances a base classifier would find it informative to label. By carefully choosing the training data, the classifier can perform well even with relatively sparse supervision. This vision has motivated a large body of work in active learning that has collectively proposed dozens of query criteria based on different theoretical or intuitive motivations, such as margin (Tong & Koller, 2002) and uncertainty-based (Kapoor et al., 2007) sampling, expected error reduction (Roy & McCallum, 2001), representative and diversity-based (Chattopadhyay et al., 2012) sampling, or combinations thereof (Hsu & Lin, 2015). It is hard to pick a clear winner all these methods, because each is based on a reasonable and appealing – but completely different – motivation; and there is no consistent winner in terms of performance across all datasets.

Rather than hand-designing a criterion and hoping that it performs well, we take a data-driven learning-based approach. We treat active learning algorithm development as a meta-learning problem and train an active learning policy represented by a neural network using deep reinforcement learning (DRL). It is natural to represent AL as a sequential decision making problem since each action (queried point) affects the context (available query points, state of the base learner) successively for the next decision. In this way the active query policy trained by RL can potentially learn a powerful and non-myopic policy. By treating the increasing accuracy of the base learner as the reward, we optimise for the actual goal: the accuracy of a classifier with a small number of labels. As the class of deep neural network (DNN) models we use includes many classic criteria as special cases, we can expect this approach should be at least as good as existing methods and likely better due to exploiting more information and non-myopic optimisation of the actual evaluation metric.

This idea of learning the best criterion within a very general function class is appealing, and other very recent research has had similar inspiration (Bachman et al., 2017). However it does not provide a general solution to AL unless the learned criterion generalises across diverse datasets/learning problems. With DRL we can likely learn an excellent query policy for any given dataset. But this is not necessarily useful alone: if we had the labels required to train the policy on a specific problem, we would not need to do AL on that problem in the first place. Thus the research question for AL

moves from "what is a good criterion?" to "how to learn a criterion that generalises?". In this paper we investigate how to train AL query criteria that generalise across tasks/datasets. Our approach is to define a DNN query criterion policy that is *parameterised by a dataset embedding*. By multi-task training of our DNN policy on a diverse batch of source tasks/datasets, the network learns how to calibrate its strategy according to the statistics of a given dataset. Specifically we are inspired by the recently proposed auxiliary network idea (Romero et al., 2017) to define a meta-network that provides parameterised domain adaptation. The meta network generates a dataset embedding and produces the weight matricies that parameterise the main policy. Besides enabling the policy to adapt to datasets with different statistics, this also means that our policy benefits from end-to-end processing of raw features while being transferable to datasets of any feature space dimensionality. Finally, unlike Woodward & Finn (2017); Bachman et al. (2017) our framework is agnostic to the base classifier. Treating the underlying learner as part of the environment to be optimised means our framework can be applied to improve the label efficiency of any existing learning architecture or algorithm.

## 2 PRELIMINARIES

**Reinforcement Learning (RL)**   In a general model-free reinforcement learning setting, an agent interacts with an environment $\mathcal{E}$ over a number of discrete time steps $t$. At each time step, the agent receives the state $s_t \in \mathcal{S}$ from environment and selects an action $a_t \in \mathcal{A}$ based on its policy $\pi(a_t|s_t)$ which is a mapping from state to action. The agent then receives a receive a new state $s_{t+1}$ and immediate reward $r_t$ from $\mathcal{E}$. The aim of RL is to maximise the return $R = \sum_{t=1}^{\infty} \gamma^{t-1} r_t$ where the return is the accumulated immediate rewards with discount factor $\gamma \in (0, 1]$. There are multiple approaches to learning the policy $\pi$ (Kober & Peters, 2009; Mnih et al., 2015). We use direct policy search based RL, which learns $\pi$ by gradient ascent on the objective function $J_\pi(\theta) = \sum_{s \in \mathcal{S}} d(s) \sum_{a \in \mathcal{A}} \pi_\theta(a|s) R$, where $d(s)$ is stationary distribution of Markov chain for $\pi_\theta$.

**Active Learning (AL)**   A dataset $\mathcal{D} = \{(\boldsymbol{x}_i, y_i)\}_{i=1}^N$ contains $N$ instances $\boldsymbol{x}_i \in \mathbb{R}^D$ and labels $y_i \in \{1, 2\}$, most or all of which are unknown in advance. In active learning, at any moment the data is split between a labelled set $\mathcal{L}$ and unlabelled set $\mathcal{U} = \mathcal{D} \setminus \mathcal{L}$ where $|\mathcal{L}| \ll |\mathcal{U}|$ and a classifier $f$ has been trained on $\mathcal{L}$ so far. In each iteration, a pool-based active learner $\tau$ selects an instance from unlabelled pool $\mathcal{U}$ to query its label $\tau : \{(\mathcal{L}, \mathcal{U}, f) \to i\}$, where $i \in \{1, \ldots, |\mathcal{U}|\}$. Then the selected instance $i$ is removed from the unlabelled set $\mathcal{U}$ and added to the labelled set $\mathcal{L}$ along with its label, and the classifier $f$ is retrained based on the updated $\mathcal{L}$.

**Connection between RL and AL**   In order to go beyond the many existing heuristic criteria, we propose to model an active learning algorithm as a neural network, and formalise discovery of the ideal criterion as a deep reinforcement learning problem. Let the state of the world $s_t$ consist of a featurisation of the dataset and the state of the base classifier $s_t = \{\mathcal{L}_t, \mathcal{U}_t, f\}$. Let an active learning criterion be a policy $\pi(a_i|s)$ where the action index $i \in \{1, \ldots, |\mathcal{U}|\}$ selects a point in the unlabelled set to query. Upon querying a point the world state is updated to $s_{t+1}$ as that point is moved from $\mathcal{U}$ to $\mathcal{L}$ and $f$ is updated as the base classifier is retrained. Assume the policy is a neural network parameterised by weights $\theta$, that selects actions as $\pi(a_i|s_t) \propto \exp^{\Phi_\theta(a_i, s_t)}$, where $i \in \{1, \ldots, |\mathcal{U}|\}$ is the index of the unlabelled instances. Finally, we define the reward of an episode to be the quantity we wish to maximise. E.g., If the budget is $N$ queries and we only care about the accuracy after the $N$th query, then we let $R = Acc_N$ where $Acc_N$ is the accuracy after the $N$th query. Alternatively, if we care about the performance during all the $N$ queries, we can use $R = \sum_{t=1}^N \gamma^{t-1} Acc_t$. (This illustrates an important advantage of the learning active learning approach: we can tune the learned criterion to suit the requirements of the AL application.) By training $\theta$ to maximise the objective $J(\theta)$ we obtain the optimal active learning policy. In interpreting AL criterion learning as a DRL problem, there is the special consideration that unlike general RL problems, each action can only be chosen once in an episode. We will achieve this by defining a fully convolutional policy network architecture where the dimensionality of the output softmax $\pi(a_i|s_t)$ can vary with $t$.

## 3 METHODS

Recall that our aim is to obtain the parameters $\theta$ of an effective dataset-agnostic active query policy $\pi_\theta(a|s)$. The two key challenges are how to learn such a policy given that: (i) the testing dataset

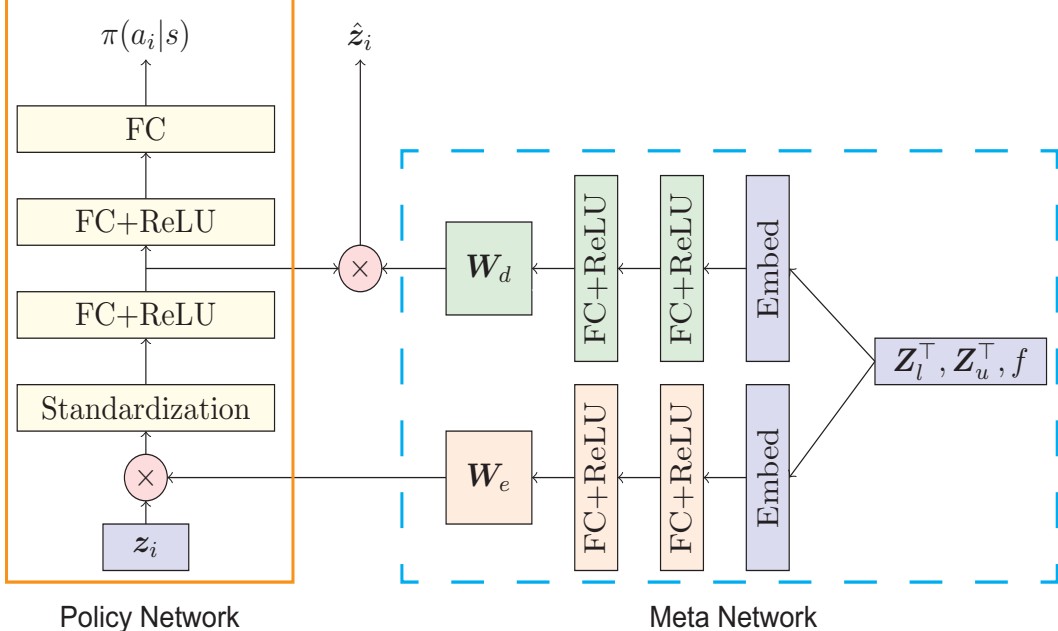

Figure 1: Policy and Meta Network architecture for deep reinforcement learning of a task-agnostic active query policy. Policy net inputs data-points $z_i$ and outputs a probability of querying them $\pi(a_i|s)$. The policy network is paramaterised by weights $\boldsymbol{W}_e$ that dynamically determined by the meta network based on an embedding of the dataset and classifier $s_t = \{\mathcal{L}_t, \mathcal{U}_t, f\}$.

statistics may be different from training dataset statistics, and moreover (ii) different datasets have different feature dimensionality $d$. This challenge is addressed by defining the overall policy $\pi_\theta(a|s)$ in terms of two sub-networks – a policy network and meta network – described as follows.

**Policy Network**     Overall the policy network $\pi$ inputs all $N$ unlabelled instance $\boldsymbol{Z}_u \in \mathbb{R}^{N \times d}$ and its output is an $N$-way softmax distribution for selecting choice of instance to query. We assume the policy models actions via the softmax $\pi(a_i|s) \propto \exp^{\Phi_{\theta_p}(\boldsymbol{W}_e^T \boldsymbol{z}_i)}$, where $\boldsymbol{z}_i \in \mathbb{R}^d$ is the $i$th unlabelled instance in $\boldsymbol{Z}_u$ and $\boldsymbol{W}_e \in \mathbb{R}^{d \times k}$ encodes the pool of instances. Although dimensionality $d$ varies by dataset, the encoding $\boldsymbol{u}_i = \boldsymbol{W}_e^T \boldsymbol{z}_i \in \mathbb{R}^k$ does not, so the rest of the policy network $\pi(a_i|s) \propto \exp^{\Phi_{\theta_p}(\boldsymbol{u}_i)}$ is independent of dataset dimension. The key is then how to obtain encoder $\boldsymbol{W}_e$ which will be provided by the meta network. Following previous work (Bachman et al., 2017; Konyushkova et al., 2017) we also allow the instances to be augmented by instance-level expert features so $\boldsymbol{Z} = [\boldsymbol{X}, \boldsymbol{\xi}(\boldsymbol{X})]$ where $\boldsymbol{X}$ are the raw instances and $\boldsymbol{\xi}(\boldsymbol{X})$ are the expert features of each raw instance.

**Meta Network**     The encoding parameters $\boldsymbol{W}_e$ of the policy network are obtained from the meta network: $\Psi_{\theta_m^e} : \{(\mathcal{L}, \mathcal{U}, f) \to \boldsymbol{W}_e; \theta_m^e\}$. The meta network inputs a featurisation of $\mathcal{L}, \mathcal{U}$ and $f$ and produces $\boldsymbol{W}_e \in \mathbb{R}^{d \times k}$ to allow the policy network to process $d$-dimensional inputs into a fixed $k$-dimnesional hidden representation. Following Romero et al. (2017) we also use the $\boldsymbol{W}_d \in \mathbb{R}^{k \times d}$ dimensional decoder $\Psi_{\theta_m^d} : \{(\mathcal{L}, \mathcal{U}, f) \to \boldsymbol{W}_d; \theta_m^d\}$to regularise this process by reconstructing the input features. The meta network synthesises these weight matricies based on dataset-embeddings of $\boldsymbol{Z}^T$ described in the following section.

### 3.1 ACHIEVING CROSS DATASET GENERALISATION

The idea of auxiliary networks to predict weights for a target network was recently used in Romero et al. (2017). There the auxiliary network inputs an embedding of $\boldsymbol{X}^T$ and predicts the weights for a main network that inputs $\boldsymbol{X}$, with the purpose of reducing the total number of parameters if $\boldsymbol{X}$ is high dimensional. In Romero et al. (2017) all the training and testing is performed on the same dataset. Here we are inspired by this idea in proposing a meta-network strategy for achieving end-

to-end learning of multiple-domains. By multi-task training on multiple datasets, the meta-network learns to generate dataset-specific weights for the policy network such that it performs effectively on all training problems and generalises well to new testing problems based on their embedding.

**Dimension Embedding Strategy**  The auxiliary meta-network requires a feature embedding that produces a fixed size description of each dimension across all datasets. The meta network takes $(\mathcal{L}, \mathcal{U}, f)$ as input, treating each feature as an example. It extracts an embedding from each input (feature) and then predicts the policy network's weights for the corresponding feature. All together, the auxiliary network predicts the weight matrix $\boldsymbol{W}_e \in \mathbb{R}^{d \times k}$, which the policy network can use to map each feature dimension to a $k$ dimensional embedding, as

$$(\boldsymbol{W}_e)_j = \Psi\Big( [\boldsymbol{e}_j^1(\boldsymbol{Z}_u^T), \boldsymbol{e}_j^1(\boldsymbol{Z}_l^T), \boldsymbol{e}_j^2([\boldsymbol{Z}_u^T, \boldsymbol{Z}_l^T], f)] \Big). \tag{1}$$

Here $e$ is a non-linear feature embedding, $j$ indexes features, selecting the $j$th embedded feature and the $j$th row of $\boldsymbol{W}_e$, and $\Psi$ is the non-linear mapping of the meta-network, which outputs a vector of dimension $k$. Similarly, the meta-network also predicts the weight matrix $\boldsymbol{W}_d$ used for auto-encoding reconstruction (Fig 1). Although $d$ is dataset dependent, the meta network generates weights for a policy network of appropriate dimensionality ($d \times k$) to the target problem. The specific embeddings used are explained next.

**Choice of Embeddings**  We use two 'representative' and 'discriminative' histogram style embeddings. The dimension-level embedding is to embed each feature dimension into a $h$ histogram. **Representative** For the representative embedding ($\boldsymbol{e}_j^1(\boldsymbol{Z}_u^T)$ and $\boldsymbol{e}_j^1(\boldsymbol{Z}_l^T)$), we encode each feature dimension as a histogram over the instances in that dimension. Specifically, we rescale the $i$th dimension features into $[0, 1]$ and divide the dimension into 10 bins. Then we count the proportion of labelled and unlabelled data for each bin. This gives a $1 \times 20$ histogram embedding for each dimension that encodes its moments. **Discriminative** ($\boldsymbol{e}_j^2([\boldsymbol{Z}_u^T, \boldsymbol{Z}_l^T], f)$) In this case we create a 2-D histogram of 10 bins per dimension. In this histogram we count the frequency of instances with feature values within each bin (as per the previous embedding) jointly with the frequency of instances with posterior values within each bin (ie, binning on the [0,1] posterior of the binary base classifier.) Finally procedure counts in a $10 \times 10$ grid, which we vectorise to $1 \times 100$. Concatenating these two embeddings we have that $[\boldsymbol{e}_j^1(\boldsymbol{Z}_u^T), \boldsymbol{e}_j^1(\boldsymbol{Z}_l^T), \boldsymbol{e}_j^2([\boldsymbol{Z}_u^T, \boldsymbol{Z}_l^T], f)]$ provides a $E = 120$ dimensional representation of each feature dimension for processing by the meta network.

**Training for Cross Dataset Generalisation**  We train policy networks and meta networks using the policy gradient method REINFORCE (Williams, 1992) to ensure that the generated policies maximise the return (active learning accuracy) with the desired reward discounting. To ensure that our pair of networks achieve the desired dataset (active learning problem) invariance, we perform multi-task training on multiple source datasets: (i) In every mini batch we sample a random subset of source datasets, and set the return to the average return over all the sampled datasets. Thus achieving a high return means the meta network has learned to synthesise suitable per-dataset weights for the policy network based on the dataset embedding, and that together they generalise across multiple tasks/datasets. (ii) To further promote cross-dataset generalisation, we apply the baseline method to standardise the return from each episode which compensates the diverse return scale across different datasets. This relative return alleviates the risk of domination by a single dataset with large return due to differing scale of accuracy increments among datasets of varying difficulty. The overall training algorithm is summarised in Alg. 1.

## 3.2 Reinforcement Learning Training and Objective Functions

The ideal active learner should query the instance that maximally improves the base learner performance. The reward that reflects the quantity we care about is therefore the increase of test split accuracy $r_t = Acc_t - Acc_{t-1}$. To optimise this quantity non-myopically, we define the return of an active learning session as $J(\theta) = \mathbb{E}[\sum_{t=1}^{\infty} \gamma^{t-1} r_t(s, \pi_\theta(\cdot, s))]$. We then use policy gradient to train the policy and meta-networks to optimise the objective $J(\theta)$.

**Auxiliary Regularisation Losses**  Besides optimising the obtained reward, we also optimise for two auxiliary regularisation losses.  **Reconstruction:** The policy network should reconstruct the unlabelled input data using $\boldsymbol{W}_d$ predicted by the meta-network (Romero et al., 2017). We optimise $A(\boldsymbol{Z}_u) = |\boldsymbol{Z}_u - \hat{\boldsymbol{Z}}_u|_F$, the mean square reconstruction error of the autoencoder. **Entropy Regularisation:** Following Mnih et al. (2016), we also prefer a policy that maintains a high-entropy

---

**Algorithm 1** Reinforcement Learning of a Transferable Query Policy

---

    **Input:**
1: **for** < each iteration > **do**                                             $\triangleright 1 \ldots 50,000$
2:     **for** < each episode > **do**                                     $\triangleright$ Collect batch
3:         Pick source dataset randomly
4:         Initialise label and unlabelled pool
5:         **for** < each time step to time T > **do**
6:             Sample action $\pi(a_i|s) \propto \exp^{\Phi_{\theta_p}(\boldsymbol{W}_e^T \boldsymbol{z}_i)}$
7:             Update the $\boldsymbol{Z}_u, \boldsymbol{Z}_l$ and base learner $f$
8:             Record the triplet $< \boldsymbol{Z}_u, a, r >$         $\triangleright$ state, action, reward
9:         **end for**
10:         Standardise episode-collected return
11:     **end for**
12:     Update Policy with standardised return
13: **end for**
14: **return** Trained Active Query Policy

---

posterior over actions so as to continue to explore and avoid pre-emptive convergence to an over-confident solution.

Integrating the main RL and two auxiliary supervised tasks together, we train both networks end-to-end. We maximise the whole objective function $F$ by reversing the sign of reconstruction loss:

$$F = J_\theta(\Phi) - \lambda_1 A_{\theta_m^d}(\boldsymbol{Z}_u) + \lambda_2 \mathcal{H}(\pi_\theta(\boldsymbol{a}|\boldsymbol{Z}_u)) \tag{2}$$

where $\theta = \{\theta_p, \theta_m^e\}$. The network (Fig. 1) trained by Eq. 2 using Alg. 1 learns to synthesise policies that are effective active query criteria (high return $J$) on any domain/dataset (synthesising domain specific network parameters via auxiliary network), adapting to the statistics of the dataset and independent of the dimensionality of the dataset.

## 4 EXPERIMENTS

### 4.1 DATASETS AND SETTINGS

**Datasets** We experiment with a diverse set of 14 datasets from UCI machine learning repository. These include *austra*, *heart*, *german*, *ILPD*, *ionospheres*, *pima*, *wdbc*, *breast*, *diabetes*, *fertility*, *fourclass*, *habermann*, *livers*, *planning*. For our main experiment, we use leave-one-out: multi-task training the policy and auxiliary network on 13 datasets, and evaluating on the held out dataset.

**Architecture** The auxiliary network for encoder has fully connected layers with of size $120, 100, 100$ ($E = 120, k = 100$) and decoder auxiliary network has analogous structure. The policy network has layers of size $N \times d$ ($N \times d$ input matrix $\boldsymbol{Z}_u$), $N \times 100$ $N \times 50$, $N \times 10$, $N \times 1$ ($N$-way output). All penultimate layers use ReLU activation. The transition of the input to first hidden layer of policy network is provided by the auxiliary network. Thereafter for efficient implementation with few parameters and to deal with the variable sized input and output, the policy network is implemented convolutionally. We convolve a $h_1 \times h_2$ sized matrix across the $N$ dimension of each $N \times h_1$ matrix shaped layer to obtain the next $N \times h_2$ layer.

**Experiment Settings** We train using Adam optimiser with initial learning rate 0.001 and hyper-parameters set to $\lambda_1 = 1$, $\alpha = \lambda_2 = 0.005$ and discount factor $\gamma = 0.99$. During RL training, we use two tricks to stabilise the policy gradient. 1) We use a relatively large batch size of 32 episodes. 2) We smooth the gradient by accumulated time-step $G_t = (1 - \alpha)G_{t-1} + \alpha g_t$ where $g_t$ is the gradient of the $a_t$ in time step $t$ and the $G_t$ is the accumulated gradient. Intuitively, the accumulated gradient $G_t$ puts more emphasis on early time step actions. We train the policy and meta network simultaneously for a fixed 50,000 iterations and perform active learning over a time horizon (budget) of 20. As base learner we explore linear SVM and RBF SVM (kernel bandwidth 0.5) with class balancing. All results shown are averages over 100 trials of training and testing datasets. **Expert Features:** To enhance the low-level feature of each instance in $\boldsymbol{X}$ we define expert features $\xi(\boldsymbol{X})$ to include distance furthest first and uncertainty as the augmented feature.

**Alternatives**    We compare our learning approach to AL with three classic approaches uncertainty/margin-based sampling (US) (Tong & Koller, 2002; Kapoor et al., 2007), furthest-first sampling (DFF) (Baram et al., 2004) and query-by-bagging (QBB) (Abe & Mamitsuka, 1998), as well as to random sampling (RAND) as a lower bound. Uncertainty sampling is a simple deterministic approach that queries the instance with minimum certainty (maximum entropy). While simple, and not the most state of the art criteria, it is consistently very competitive with more sophisticated criteria and more robust in the sense of hardly ever being a very poor criteria. As a representative more sophisticated approach, we compare with QUIRE (Huang et al., 2010) and as a recent (within-dataset) learning based approach, we compare ALBL (Hsu & Lin, 2015). We denote our method meta-learned policy for general active learning (MLP-GAL). As a related alternative we propose SingleRL. This is our RL approach, but without the meta-network, so a single model is learned over all datasets. Without the meta-network it can only use expert features $\xi(\boldsymbol{X})$ so that dimensionality is fixed over datasets. To give SingleRL an advantage we concatenate some extra global features to the input space[1]. This method can also be seen as a version of one of the few state of the art learning-based alternatives (Konyushkova et al., 2017). But upgraded in that we learn it with reinforcement learning instead of the more myopic supervised learning used in (Konyushkova et al., 2017).

## 4.2    RESULTS

**Multi-Task Training Evaluation**    We first verify that it is indeed possible to learn a single policy that generalises across multiple training datasets with Linear SVM. In our leave-one-out setting, this means generalising across 13 datasets simultaneously. Each result in the MLP-GAL (Tr) column of Table 1 is an average across the 13 combinations in which the corresponding dataset occurs in multi-task training. We can see that MLP-GAL learns an effective criterion that outperforms the competitors. There is potential for overfitting as the policy has seen each dataset during training (datasets randomly selected in minibatches). However it is interesting that it works because it shows that it is possible to learn a *single* query policy that performs well on such a diverse set of datasets.

**Cross-Task Generalisation**    In the next experiment we apply our multi-task trained method to held out datasets. In the leave-one-out setting, this means that each row in Table 1 represents a testing set, and the MLP-GAL (Te) result is the performance on this test set after training on all 13 other datasets. Our MLP-GAL outperforms alternatives in both average performance and number of wins. SingleRL is generally also effective compared to prior methods, showing the efficacy of training a policy with RL. However it does not benefit from a meta network, so is not as effective as our MLP-GAL. From the table it is also interesting to see that while sophisticated methods such as QUIRE sometimes perform very well, they also often perform very badly – even worse than random. Meanwhile the simple and classic uncertainty-sampling and QBB methods perform consistently well. Their robustness is the reason for their continued use in practice despite their age and simplicity. This dichotomy illustrates the challenge in building sophisticated AL algorithms that generalise to datasets that they were not engineered on. In contrast, although our approach MLP-GAL (Te) has not seen these datasets during training, it performs consistently well due to adapting to each dataset via the meta-network. Fig 2(a) shows the resulting active learning curve for an example dataset.

**Application to RBF SVM learner**    An advantage of our approach compared to related methods such as Bachman et al. (2017); Woodward & Finn (2017) is that it treats the base learner as part of the environment to be optimised against rather than tying the user to a particular learner. Applying our method to RBF SVM base learner, we can see that the results in Table 2 are similar to linear SVM (expected given the difficulty of learning a non-linear model in a budget of 20 points). However our learning-based approach is again consistently high performing and effective overall – it is able to learn a policy customised for this new type of base learner.

**Dependence on Number of Training Domains**    We next investigate how performance depends on the number of training domains. We train MLP-GAL with an increasing number of source datasets – 1, 4, 7 (multiple splits each); or 13 (13 split LOO setting). Then we compute the average performance over all training and all testing domains, in all of their multiple occurrences across the splits. From the results in Fig 2(b) we see that the training performance becomes worse when doing a higher-way multi-task training. This is intuitive: it becomes harder to overfit to more datasets

---

[1]Variance of classifier weight, proportion of labelled pos/neg instances, proportion of predicted unlabelled pos/neg instances', proportion of budget used (Konyushkova et al., 2017)

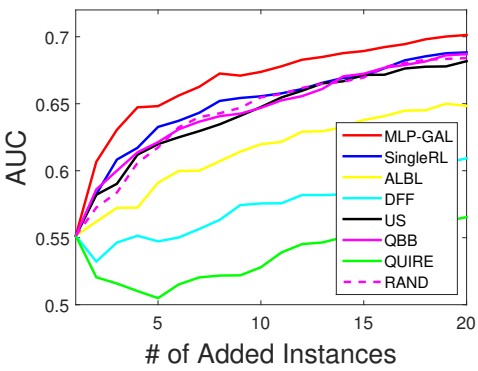
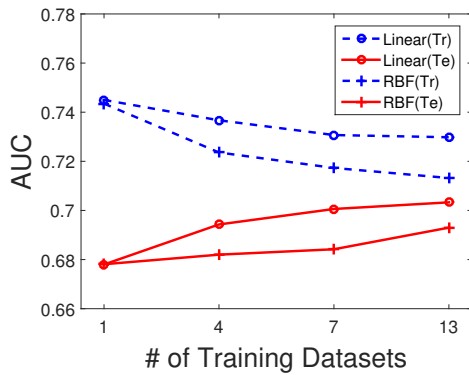

(a) Illustrative active learning curves from evaluating our learned policy on held out UCI dataset diabetes.

(b) Cross-dataset generalisation. Average performance (AUC) of MLP-GAL over all training and testing sets as a function of the number of training domains.

Figure 2: Further Analysis

simultaneously. Meanwhile testing performance improves, demonstrating that the model learns to generalise better to held out problems when forced to learn on a greater diversity of source datasets.

## 5 RELATED WORK

**Active Learning by Learning**    A few papers have very recently appeared that also approach finding an AL criterion as a learning problem. Konyushkova et al. (2017) proposes to learn a criterion based on a vector of expert features (e.g., classifier confidence, label imbalance). However by using expert features, this misses the chance to learn the representation from raw features as in our approach; and by using supervised rather than reinforcement learning to train the policy, it is not optimally non-myopic. Bachman et al. (2017) and Woodward & Finn (2017) use RL to train a single model that provides both the base classifier and the active learner. This tight integration has the drawback that the frameworks are constrained to a specific base learner, so cannot be used to improve the training of an arbitrary base learner as per our framework. More importantly, while these methods learn effective non-myopic policies, they are trained and tested on different classes within the same dataset, so the generalisation challenge and evaluation is minimal. There is no mechanism to ensure effective transfer across datasets of different statics or to allow any transfer at all across datasets of different dimensionality.

**Active Learning Ensembles**    Different AL algorithms perform well on different datasets, or at different learning stages. For this reason studies have proposed heuristics to switch criteria from early to late stage learning (Donmez et al., 2007; Baram et al., 2004), or use multi-armed bandit (MAB) approaches to estimate the best criterion for a given dataset within an ensemble (Hsu & Lin, 2015). But aside from being myopic, MAB learners do not learn transferrable knowledge: They perform all their learning within a single rollout, and their need to explore/learn online is fundamentally at odds with active learning. Chu & Lin (2016) ameliorate this somewhat with regularisation, but still need dataset-specific learning. Our approach can address these issues: Besides non-myopic policy learning with RL, a DNN has capacity to encode multiple criteria and apply different ones at different stages of learning. By learning a meta-policy that paramaterises a dataset-specific policy, it customises the overall active learning strategy to the target dataset; thus transferring knowledge for immediate efficacy on a new dataset without dataset specific learning.

**Domain Generalisation and Adaptation**    Our task-agnostic AL goal is related to Domain Generalisaton (DG) (Muandet et al., 2013) and Domain Adpatation (DA) (Ganin & Lempitsky, 2015) in supervised learning in that we would like to train on one dataset and perform well when testing on another dataset. Our framework has aspects of DG (multi-task training to increase generality) and DA (adapting to target data, via dataset embedding meta network) methods. But we are not aware of any dataset embedding approaches to achieving DA within supervised learning.

Table 1: Comparison of active learning algorithms, leave one dataset out setting. Linear SVM base learner. AUC averages (%) over 100 trials (and 13 training occurrences for MLP-GAL (Tr)).

|  | MLP-GAL (Tr) | MLP-GAL (Te) | SingleRL (Te) | Entropy | DFF | RAND | ALBL | QUIRE | QBB |
|---|---|---|---|---|---|---|---|---|---|
| austra | 80.14 | 77.49 | 75.72 | 78.24 | 75.63 | 75.87 | 75.31 | 64.46 | **78.58** |
| breast | 96.67 | 95.38 | 94.78 | 95.41 | **95.76** | 94.71 | 95.67 | 95.60 | 95.73 |
| diabetes | 67.53 | **66.65** | 64.78 | 64.18 | 57.31 | 64.05 | 61.35 | 53.75 | 64.46 |
| fertility | 78.26 | 73.59 | **77.86** | 75.79 | 70.44 | 71.28 | 66.92 | 54.93 | 73.87 |
| fourclass | 74.79 | **72.02** | 71.83 | 69.55 | 71.26 | 69.08 | 68.69 | 64.48 | 70.81 |
| haberman | 67.31 | 64.47 | **64.91** | 60.16 | 60.26 | 57.40 | 52.49 | 45.89 | 60.58 |
| heart | 76.68 | 72.46 | 72.84 | 73.38 | **73.99** | 73.06 | 71.78 | 67.07 | 73.36 |
| german | 68.01 | **65.89** | 63.35 | 63.34 | 61.78 | 62.77 | 61.74 | 51.82 | 64.16 |
| ILPD | 62.48 | 58.41 | **61.08** | 57.60 | 50.97 | 57.62 | 52.91 | 48.57 | 56.77 |
| ionospheres | 74.96 | 67.31 | 69.78 | **70.47** | 59.64 | 69.81 | 68.44 | 57.84 | 70.40 |
| liver | 55.66 | 55.41 | **55.62** | 53.45 | 52.87 | 52.87 | 51.25 | 48.11 | 52.13 |
| pima | 67.64 | **66.89** | 64.67 | 64.18 | 57.31 | 63.69 | 61.27 | 53.75 | 64.24 |
| planning | 60.74 | **58.12** | 56.75 | 55.09 | 52.77 | 54.17 | 49.46 | 39.90 | 55.43 |
| wdbc | 90.90 | 90.57 | 88.72 | **90.93** | 87.55 | 88.52 | 88.41 | 82.17 | 90.68 |
| **Avg** | 72.98 | **70.33** | 70.19 | 69.41 | 66.25 | 68.21 | 66.12 | 59.17 | 69.37 |
| **Num Wins** | - | **5** | 4 | 2 | 2 | 0 | 0 | 0 | 1 |

Table 2: Comparison of active learning algorithms, leave one dataset out setting. RBF SVM base learner. AUC averages (%) over 100 trials (and 13 training occurrences for MLP-GAL (Tr)).

|  | MLP-GAL (Tr) | MLP-GAL (Te) | SingleRL (Te) | Ent | DFF | RAND | ALBL | QUIRE | QBB |
|---|---|---|---|---|---|---|---|---|---|
| austra | 80.84 | 79.14 | 76.35 | **79.36** | 77.15 | 78.47 | 76.57 | 68.98 | 78.83 |
| breast | 96.25 | 95.36 | 95.46 | 95.40 | 95.78 | 95.14 | **95.92** | 95.21 | 95.43 |
| diabetes | 66.55 | **64.28** | 62.52 | 62.59 | 59.81 | 62.7 | 59.09 | 58.48 | 61.98 |
| fertility | 80.83 | 77.8 2 | 75.75 | **79.49** | 75.81 | 75.21 | 73.55 | 64.67 | 76.83 |
| fourclass | 71.66 | **69.78** | 66.41 | 66.88 | 68.62 | 66.29 | 66.43 | 64.85 | 63.35 |
| haberman | 58.01 | 56.42 | 53.88 | 56.60 | 58.67 | 53.58 | 64.44 | 61.83 | **64.97** |
| heart | 77.47 | 73.93 | 71.87 | 73.63 | **74.05** | 72.27 | 72.57 | 68.98 | 72.95 |
| german | 67.94 | **65.78** | 64.18 | 65.01 | 65.6 | 63.26 | 57.70 | 55.57 | 53.96 |
| ILPD | 54.5 | **53.54** | 51.04 | 50.99 | 47.29 | 52.30 | 47.62 | 46.54 | 51.15 |
| ionospheres | 80.94 | 76.14 | 72.87 | **77.76** | 61.49 | 75.17 | 75.00 | 61.72 | 77.18 |
| liver | 51.91 | 49.95 | 50.76 | 50.31 | **51.04** | 50.21 | 47.60 | 46.75 | 50.27 |
| pima | 66.60 | **63.58** | 63.15 | 62.59 | 59.81 | 63.01 | 58.13 | 58.48 | 61.74 |
| planning | 53.05 | **53.55** | 52.61 | 49.95 | 50.07 | 50.99 | 47.10 | 41.68 | 50.49 |
| wdbc | 91.97 | 90.93 | 90.04 | **91.54** | 89.37 | 90.24 | 89.52 | 88.14 | 90.34 |
| **Avg** | 71.32 | **69.30** | 67.64 | 68.72 | 66.75 | 67.77 | 66.52 | 62.99 | 67.82 |
| **Num Wins** | - | **6** | 0 | 4 | 2 | 0 | 1 | 0 | 1 |

**Related Methods** Models that predict the parameters of other models are increasingly widely used (Ha et al., 2017). In robot control, such 'contextual' or 'paramaterised' policies are used to solve related tasks such as reaching to different targets (Kupcsik et al., 2013). Romero et al. (2017) used auxiliary networks for parameter reduction when training and testing on one dataset.

## 6 DISCUSSION

We have proposed a learning-based perspective on the problem of active query criteria design. Such learning-based algorithm design elegantly obtains AL models by optimising the ultimate goal of classification performance with few labels. However aside from the widely-shared questions of good network architecture and RL training algorithms, the key challenge is then whether general enough policies can be learned to be widely useful in different applications, rather than requiring dataset-specific training which contradicts the motivation of AL. Our key contribution is to provide the first solution to this issues through multi-task training of a meta-network that synthesises dataset-specific active query policies.

Our study thus far has the main limitation that we have only evaluated our method on a binary base classifier (binary assumption shared by Konyushkova et al. (2017)). In future work we would like to evaluate our method on deep multi-class classifiers by designing embeddings which can represent the state of such learners, as well as explore application to the stream-based AL setting.

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

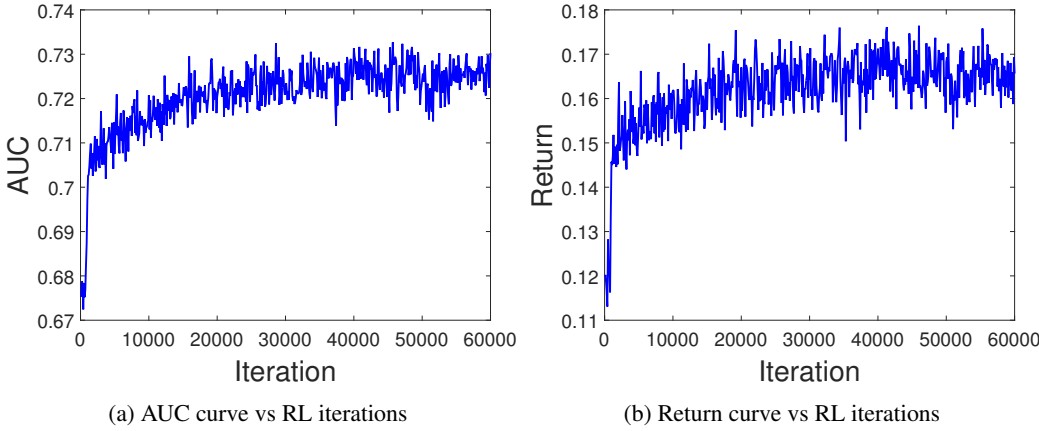

(a) AUC curve vs RL iterations        (b) Return curve vs RL iterations

Figure 3: Convergence of active learning policy during training. Austra dataset.

# 7 APPENDIX

## 7.1 CONVERGENCE PROCESS DURING LEARNING

To illustrate the stability of our reinforcement learning approach to active query policy learning, we show the convergence process during training in Fig. 3 for an example dataset *austra*. Fig. 3(a) plots the classifier AUC after after all 20 active queries made by the MLP-GAL policy, measured after different numbers of minibatch iterations of training MLP-GAL with RL. Correspondingly Fig. 3(b) plots the return (sum of discounted reward over the entire active learning rollout). We can see that both quantities increase rapidly and have stabilised after about 50,000 iterations.

## 7.2 DEPENDENCE ON CHOICE OF TRAINING DOMAINS

In our learning-based approach to AL policy generation, it is intuitive that testing performance would be dependent on the relatedness of the training datasets used. At one extreme if we train and test on the same dataset, we obtain great performance (MLP-GAL (Tr) in Tables 1 and 2). At the other extreme if we train on a *single* source dataset very unrelated to our test set, then performance may be poor. Our aim is to achieve robustness to choice of source dataset via the domain generalisation provided by dataset embeddings and our multi-task trained meta-network. To study this we perform 1-way, 4-way, 7-way and 13-way multi-task training with testing on the remaining datasets – and repeat this for multiple splits (cf. Fig 2(b)). Note that each split is designed with an inclusive structure as number of datasets increases: So for a given split, the single task (1-way) training dataset is a subset of the training datasets in the 4-way setting, which is a subset of 7-way, which is in turn a subset of the 13-way (leave-one-out) setting. We then compute the mean and standard deviation (SD) in performance over the occurrences of each dataset across all the multiple splits. The number of wins among the MLP-GAL variants evaluated here is also reported. Note that there is no training SD for single-task training (column Train-1) since there is only one way to perform single task training; and there is no testing SD after 13-task leave-one-out training (column Test-13) since there is only one possible split that leaves a single dataset out.

From the results in Tables 3 (Linear SVM) and 4 (RBF SVM) we can observe the following: (i) As the number of source datasets increases, testing performance increases – on average and for the majority of testing sets. (ii) The *testing* SD for both Linear and RBF SVM decreases with number of training domains used. (iii) For RBF SVM the *training* SD also decreases slightly with number of training domains. This suggests that being exposed to more training datasets facilitates finding a more consistent optimal policy during learning. (iv) The high testing SD of single source training (column Test-1) indicates unsurprisingly that it is sensitive to the specific choice of source dataset. A relevant source dataset can result in high performance and an irrelevant source dataset can produce low performance. So the model would work well only if a relevant source set was carefully picked. However the average performance of multi-source training (column Test-13, Average) is often similar to the upper bound achieved by single source training given its SD (column Test-1,

Table 3: MLP-GAL training and testing performance as a function of number of training datasets. AUC average and standard deviation. Linear SVM base classifier. Each dataset is evaluated both as train and test during cross validation.

| | Train Performance | | | | Test Performance | | | |
|---|---|---|---|---|---|---|---|---|
| Num Train Sets: | 1 | 4 | 7 | 13 | 1 | 4 | 7 | 13 |
| austra | **81.49** | 80.81 ± 0.57 | 80.56 ± 0.6 | 80.15 ± 0.65 | 72.78 ± 2.99 | 72.9 ± 1.97 | 74.07 ± 3.11 | **77.49** |
| breast | **96.94** | 96.85 ± 0.12 | 96.76 ± 0.16 | 96.67 ± 0.13 | 94.55 ± 1 | 95.36 ± 0.66 | 95.31 ± 0.36 | **95.38** |
| diabetes | **69.23** | 67.25 ± 0.29 | 67.22 ± 0.48 | 67.53 ± 0.45 | 63.05 ± 2.63 | 65.03 ± 1.91 | 65.56 ± 1.16 | **66.65** |
| fertility | **79.9** | 79.2 ± 0.3 | 78.38 ± 0.36 | 78.26 ± 0.61 | 72.9 ± 1.91 | 73.86 ± 1.44 | **74.91** ± 2.37 | 73.59 |
| fourclass | **76.03** | 75.36 ± 0.48 | 75.08 ± 0.37 | 74.79 ± 0.47 | 69.15 ± 2.15 | 71.24 ± 2.69 | **73.02** ± 0.78 | 72.02 |
| haberman | **71.33** | 68.06 ± 0.74 | 66.91 ± 0.84 | 67.31 ± 0.62 | 59.28 ± 3.57 | 62 ± 2.52 | **64.97** ± 0.54 | 64.47 |
| heart | **80.3** | 78.46 ± 1.19 | 77.48 ± 0.56 | 76.68 ± 0.74 | 70.38 ± 2.84 | **72.5** ± 1.79 | 72.35 ± 0.92 | 72.46 |
| german | 68.55 | **68.6** ± 0.36 | 68.1 ± 0.16 | 68.01 ± 0.33 | 64.05 ± 2.04 | 64.44 ± 0.99 | 65 ± 1.67 | **65.89** |
| ILPD | **65.26** | 64.01 ± 1.04 | 62.97 ± 0.82 | 62.48 ± 1.07 | 56.17 ± 2.73 | 58.37 ± 1.5 | 58.11 ± 1.56 | **58.41** |
| ionospheres | 75.29 | **75.8** ± 1.68 | 75.21 ± 1.06 | 74.96 ± 0.78 | 68.04 ± 3.85 | **70.27** ± 1.95 | 70.12 ± 1.57 | 67.31 |
| liver | 54.88 | **56.59** ± 0.41 | 56.04 ± 0.35 | 55.66 ± 0.34 | 54.37 ± 1.07 | 54.82 ± 0.46 | 54.86 ± 0.27 | **55.41** |
| pima | **69.77** | 67.83 ± 0.31 | 66.78 ± 0.65 | 67.64 ± 0.6 | 63 ± 2.59 | 65.15 ± 1.76 | 66.2 ± 1.28 | **66.89** |
| planning | **62.61** | 61.37 ± 0.51 | 60.71 ± 0.79 | 60.74 ± 0.98 | 54.9 ± 3.14 | 57.28 ± 3.07 | 57.23 ± 1.31 | **58.12** |
| wdbc | **91.4** | 91.25 ± 0.19 | 90.78 ± 0.58 | 90.9 ± 0.25 | 86.6 ± 2.49 | 88.76 ± 0.92 | 89.16 ± 0.91 | **90.57** |
| Average | **74.5** | 73.67 ± 0.59 | 73.07 ± 0.55 | 72.98 ± 0.57 | 67.8 ± 2.5 | 69.43 ± 1.69 | 70.06 ± 1.27 | **70.33** |
| Num Wins | **11** | 3 | 0 | 0 | 0 | 2 | 3 | **9** |

Table 4: MLP-GAL training and testing performance as a function of number of training datasets. AUC average and standard deviation. RBF SVM base classifier. Each dataset is evaluated both as train and test during cross validation.

| | Train Performance | | | | Test Performance | | | |
|---|---|---|---|---|---|---|---|---|
| Num Train Sets: | 1 | 4 | 7 | 13 | 1 | 4 | 7 | 13 |
| austra | **81.84** | 81.73 ± 0.54 | 80.99 ± 0.31 | 80.84 ± 0.34 | 76.47 ± 2.01 | 76.01 ± 0.82 | 76.5 ± 2.71 | **79.14** |
| breast | 95.94 | 96.08 ± 0.17 | 96.27 ± 0.19 | **96.25** ± 0.24 | 94.74 ± 0.89 | 95.52 ± 0.32 | **95.68** ± 0.27 | 95.36 |
| diabetes | **70.37** | 66.96 ± 1.26 | 66.86 ± 0.96 | 66.55 ± 1.05 | 63.57 ± 2.98 | 63.8 ± 3.13 | **64.67** ± 2.2 | 64.28 |
| fertility | 81.7 | **81.71** ± 0.66 | 81.44 ± 0.89 | 80.83 ± 0.58 | 75.91 ± 2.8 | 76.74 ± 1.88 | **77.92** ± 0.61 | 77.82 |
| fourclass | 73.9 | 71.49 ± 0.65 | 70.73 ± 0.66 | 71.66 ± 0.66 | 66.28 ± 1.61 | 67.11 ± 1.68 | 67.81 ± 1.32 | **69.78** |
| haberman | **65.95** | 61.95 ± 1.79 | 60.41 ± 1.77 | 58.01 ± 1.56 | 54.97 ± 2.24 | 56.37 ± 2.26 | 54.87 ± 1.33 | **56.42** |
| heart | **79.7** | 79.09 ± 0.93 | 77.9 ± 0.82 | 77.47 ± 0.68 | 72.26 ± 2.09 | 73.76 ± 1.85 | 73.58 ± 1.56 | **73.93** |
| german | **70.23** | 68.73 ± 0.66 | 68.1 ± 1.34 | 67.94 ± 0.36 | 64.63 ± 3.91 | 64.14 ± 2.29 | 65.27 ± 1.35 | **65.78** |
| ILPD | **62.1** | 57.89 ± 2.87 | 55.1 ± 0.89 | 54.5 ± 0.55 | 52.45 ± 2.82 | 50.94 ± 2.2 | 50.54 ± 1.81 | **53.54** |
| ionospheres | 80.81 | **81.5** ± 0.36 | 81.67 ± 0.46 | 80.94 ± 0.55 | 73.54 ± 3.49 | 74.97 ± 2.4 | **76.87** ± 3.04 | 76.14 |
| liver | **56.01** | 51.81 ± 1.18 | 52.26 ± 0.72 | 51.91 ± 0.85 | 50.79 ± 0.79 | **50.96** ± 1.06 | **50.96** ± 0.57 | 49.95 |
| pima | **71.56** | 67.73 ± 0.83 | 66.57 ± 1.7 | 66.6 ± 1.24 | 63.36 ± 2.82 | 63.59 ± 2.32 | **65.31** ± 1.63 | 63.58 |
| planning | **57.87** | 54.2 ± 2.15 | 53.38 ± 0.58 | 53.05 ± 0.89 | 51.48 ± 1.26 | 52.02 ± 1.07 | 52.13 ± 0.63 | **53.55** |
| wdbc | **92.92** | 92.2 ± 0.3 | 92.31 ± 0.7 | 91.97 ± 0.28 | 88.87 ± 1.17 | 88.86 ± 0.97 | 90.31 ± 0.53 | **90.93** |
| Average | **74.35** | 72.36 ± 1.03 | 71.71 ± 0.86 | 71.32 ± 0.7 | 67.81 ± 2.21 | 68.2 ± 1.73 | 68.74 ± 1.4 | **69.3** |
| Num Wins | **11** | 2 | 0 | 1 | 0 | 1 | 6 | **8** |

Average+SD). EG: For Linear SVM performance in an optimistic scenario (assume relevant source data is specifically selected) is $67.8 + 2.5 = 70.3\%$ (Test-1) which is comparable to the average scenario of $70.33\%$ (Test-13) for 13-way multi-dataset training. This suggests that careful choice of a specific relevant training set is not crucial for MLP-GAL. Multi-source training on many source datasets is adequate, and our model can generalise to the new dataset via its embedding. Improving performance further by automatically determining relevant source datasets to use is an open question for potential future improvement of this work, as it is similarly an open question in transfer learning more generally.

