# OpenReview forum: "Meta-Learning Transferable Active Learning Policies by Deep Reinforcement Learning"
_ICLR.cc/2018/Conference — Reject_

### Official Review · AnonReviewer3 · 2017-11-25
**This is an exciting paper on an important topic (active learning), but it suffers from a weak empirical evaluation.**

**Rating:** 7
**Confidence:** 4

**Review:**

This reviewer has found the proposed approach quite compelling, but the empirical validation requires significant improvements:
1) you should include in your comparison Query-by- Bagging & Boosting, which are two of the best out-of-the-box active learning strategies
2) in your empirical validation you have (arbitrarily) split the 14 datasets in 7 training and testing ones, but many questions are still unanswered:
 -  would any 7-7 split work just as well (ie, cross-validate over the 14 domains)
 - do you what happens if you train on 1, 2, 3, 8, 10, or 13 domains? are the results significantly different?

OTHER COMMENTS:
- p3: both images in Figure 1 are labeled Figure 1.a
- p3: typo "theis" --> "this"

Abe & Mamitsuksa (ICML-1998). Query Learning Strategies Using Boosting and Bagging.

---

> ### Author Response · Authors · 2018-01-05
> **Experiments of 'Cross-validation' and 'dependence on number of train datasets' added**
>
> Thanks for the comments.
>
> QBB: We chose the Query-by-Bagging variant to compare as it had slightly better performance in the original QBB paper. We now show QBB results in the empirical evaluation. We can see that QBB is competitive but does not outperform our learned method.
>
> Cross-validation and dependence on number of train datasets: We agree these are interesting and important points to investigate. To improve our experiments accordingly, we have now: (1) cross-validated, treating all datasets as train/test in turn. (2) Repeated this for multiple train/test splits including 13/1 (leave-one-out), 7/7, 4/10 and 1/13 (single source training).
>
> Cross-validation: The main results in Tab 1 (Lin SVM) and Tab 2 (RBF SVM) are now based on 13/1 leave-one-dataset cross-validation out. We see that the conclusions still hold: MLP-GAL outperforms alternatives on average, when testing across all 14 of the datasets.
>
> Dependence on # of source datasets: This is now reported in Fig 2(b). We can see that increasing from 1 up to 13 train sets reduces the training performance (harder to simultaneously overfit to a larger suite of training datasets). Simultaneously it increases the testing performance (being forced to fit a wider suite of training data forces the policy to represent more dataset-agnostic knowledge, and thus improves generalisation). See also the new Appendix Sec 7.2 for further discussion.
>
> Typos: Thanks. Now fixed.

---

### Official Review · AnonReviewer2 · 2017-11-27
**A novel meta-learning way to do active learning, slightly complicated embedding strategy, needs more evidence to show if it'll generalise to more challenging problems.**

**Rating:** 6
**Confidence:** 3

**Review:**

The approach solves an important problem as getting labelled data is hard. The focus is on the key aspect, which is generalisation across heteregeneous data. The novel idea is the dataset embedding so that their RL policy can be trained to work across diverse datasets.

Pros:
1. The approach performs well against all the baselines, and also achieves good cross-task generalisation in the tasks they evaluated on.
2. In particular, they alsoevaluated on test datasets with fairly different statistics from the training datasets, which isnt very common in most meta-learning papers today, so it’s encouraging that the method works in that regime.

Cons:
1. The embedding strategy, especially the representative and discriminative histograms, is complicated. It is unclear if the strategy is general enough to work on harder problems / larger datasets, or with higher dimensional data like images. More evidence in the paper for why it would work on harder problems would be great.
2. The policy network would have to output a probability for each datapoint in the dataset U, which could be fairly large, thus the method is computationally much more expensive than random sampling. A section devoted to showing what practical problems could be potentially solved by this method would be useful.
3. It is unclear to me if the results in table 3 and 4 are achieved by retraining from scratch with an RBF SVM, or by freezing the policy network trained on a linear SVM and directly evaluating it with a RBF SVM base learner.

Significance/Conclusion: The idea of meta-learning or learning to learn is fairly common now. While they do show good performance, it’s unclear if the specific embedding strategy suggested in this paper will generalise to harder tasks.

Comments: There’s lots of typos, please proof read to improve the paper.

Revision: I thank the authors for the updates and addressing some of my concerns. I agree the computational budget makes sense for cross data transfer, however the embedding strategy and lack of larger experiments makes it unclear if it'll generalise to harder tasks. I update my review to 6.

---

> ### Author Response · Authors · 2018-01-05
> **Transferring to bigger computer vision problems is important and related experiments would be added in future edition**
>
> Thanks for the feedback.
>
> Computational Expense: We agree our model is more expensive than random sampling at run-time, as are all other non-trivial active learners (Entropy/US, DFF, QUIRE, QBB, ALBL, etc). The standard cost bracket of most active learning methods is O(ND) each iteration for N instances and D dimensions. Ours is in this bracket along with the pervasive Entropy/US, and thus we can address problems of the same size as any standard active learner. DFF, for example, is a qualitatively more costly O(N^2 D), which makes it non-scalable to large problems.
>
> Training our model is fairly costly, but by training a model capable of cross-dataset transfer, the idea is that retraining is rarely required.
>
> Linear and RBF SVM: In the RBF experiment table (Tab 3, 4 in submission. Tab 2 in revised paper), the results are based on completely retraining the policy network using a RBF-base learner. Since we did not meta-train for base-learner invariance, we would not expect it to generalise across changes of base learner. Meta-training for base learner invariance is an interesting avenue for future work.
>
> Bigger Problems: We agree this is an exciting major test for going forward in this line of research. Given the other requested experiments, which required all our GPUs, we did not have time to address this in the available period. We think that the existing experiments, particularly with the revised updates already validates the basic concept. So we leave evaluation of bigger computer vision problems for future work.
>
> Typos: Sorry. We have proofread the paper and corrected them.

---

### Official Review · AnonReviewer1 · 2017-11-29
**nice idea, needing more experiments**

**Rating:** 6
**Confidence:** 4

**Review:**

Overview

The authors propose a reinforcement learning approach to learn a general active query policy from multiple heterogeneous datasets. The reinforcement learning part is based on a policy network, which selects the data instance to be labeled next. They use meta-learning on feature histograms to embed heterogeneous datasets into a fixed dimensional representation. The authors argue that policy-based reinforcement learning allows learning the criteria of active learning non-myopically. The experiments show the proposed approach is effective on 14 UCI datasets.

strength

* The paper is mostly clear and easy to follow.
* The overall idea is interesting and has many potentials.
* The experimental results are promising on multiple datasets.
* There are thorough discussion with related works.

weakness

* The graph in p.3 don't show the architecture of the network clearly.
* The motivation of using feature histograms as embedding is not clear.
* The description of the 2-D histogram on p.4 is not clear. The term "posterior value" sounds ambiguous.
* The experiment sets a fixed budget of only 20 instances, which seems to be rather few in some active learning scenarios, especially for non-linear learners. Also, the experiments takes a fixed 20K iterations for training, and the convergence status (e.g. whether the accumulated gradient has stabilized the policy) is not clear.
* Are there particular reasons in using policy learning instead of other reinforcement learning approaches?
* The term A(Z) in the objective function can be more clearly described.
* While many loosely-related works were surveyed, it is not clear why literally none of them were compared. There is thus no evidence on whether a myopic bandit learner (say, Chu and Lin's work) is really worse than the RL policy. There is also no evidence on whether adaptive learning on the fly is needed or not.
* In Equation 2, should there be a balancing parameter for the reconstruction loss?
* Some typos
    - page 4: some duplicate words in discriminative embedding session
    - page 4: auxliary -> auxiliary
    - page 7: tescting -> testing

---

> ### Author Response · Authors · 2018-01-05
> **Detailed Reply to the concerns**
>
> Thanks for the comments.
>
> Histogram Embedding Motivation: The general idea of a histogram-based embedding was inspired by Romero ICLR'17. We customised it for application to AL here. For example, the representative embedding encodes information about the spread of the dataset itself (unlabelled component) and the spread of the queries so far (labelled component). While the discriminative embedding  contains information about how the current leaner certainty varies with the position of an instance within the spread of the dataset (through encoding posterior certainty jointly with position).
>
> Convergence: Yes the policy is stabilising. This is now illustrated in Appendix Sec 7.1, Fig 3.
>
> Fixed budget: AL is mainly of interest in settings when a small budget must be carefully spent. We agree extending to budgets of 100 or more is within the interesting range. But we leave this for future work as we didn't have time to run these experiments yet.
>
> 2D histogram: To clarify: Each feature dimension of input is encoded by a joint histogram counting: (1) The frequency of instances with a value of that feature within each bin and (2) The frequency of instances with a given posterior probability according to the base classifier so far (IE: binning on the base classifier's probability value between [0,1] that the given instance is class +1 vs class -1).  We clarified this in Sec 3.1.
>
> Explain A(Z): This is the MSE of reconstruction of the instances by the autoencoder. Now clarified in Sec 3.2.
>
> Quantitative Comparison. Myopic/ALBL: The reviewer mentioned they would like evidence of RL policy benefit vs myopic bandit learner. We would like to reiterate that the mentioned point here is only one out of two contributions here. We perform non-myopic learning with RL, but the second contribution is the learning of cross-dataset generalisation which is not attempted at all in prior work such as Chu and Lin's ALBL. We have now added ALBL to the experimental comparison, (as well as QBB suggested by R3). We can see that MLP-GAL is clearly better than ALBL in the updated Tables 1 and 2. To understand why: Recall that ALBL performs bandit-based learning within-dataset without knowledge transfer. Its underlying bandit learner is designed for asymptotic performance. In the few-shot scenario of active learning, it suffers critically from the fundamental explore/exploit dilemma. ALBL must use the limited active queries to do exploration for learning. By the time it has explored enough to define a good learner, it has a small active query budget remaining to exploit this knowledge. This is particularly detrimental in our tough setting of only 20 queries per dataset. In contrast by doing cross-dataset transfer, our MLP-GAL completely avoids this limitation. It gets all its learning done on different source datasets and can go straight to "exploiting" when tested on a novel dataset.
>
> Quantitative comparison. Adaptive learning on the Fly: We assume the reviewer is referring to the meta-network in this question. The relevant comparison is then between MLP-GAL and Single-RL which is trained in a similar multi-task way, but excludes the meta-network that helps the policy adapt to new target datasets. We can see that MLP-GAL outperform Single-RL in the results (Tab 1, and more clearly in Tab 2). This is despite that the latter adds significantly more expert features in order to make it approximate a non-myopic RL-upgraded version of the expert feature-based supervised learner in Konyushkova et al, NIPS, 2017.
>
> RL Approach chosen: For simplicity we chose PG, as the simplest classic direct policy search method. More advanced methods (actor-critic, TRPO, etc) could be potentially used to further improve performance. We tried DQN-based Q-learning and it did not work as well due to being hard to stabilise the training. We believe Q-learning should also work in principle but may be more sensitive to tuning hyperparameters for good stability.
>
> Eq 2: Yes there a potential parameter here that could be tuned to improve performance, but we did not use one. So the weight value is 1. Now clarified.
>
> Typos: Thanks. Corrected.

---

> > ### Comment · AnonReviewer1 · 2018-01-19
> > **about myopic/ALBL**
> >
> > Thanks for addressing some of the concerns. The following sentence, however, is factually inaccurate and deserves clarification.
> >
> > "learning of cross-dataset generalisation which is not attempted at all in prior work such as Chu and Lin's ALBL"
> >
> > I assume that the authors are talking about Hsu and Lin's ALBL, which did not attempt for cross-dataset generalisation. Nevertheless, the other work that is being confused here, Chu and Lin (2016)'s Transfer LSA, improves over ALBL with cross-dataset generalisation, albeit in a sequential setting with a bandit learner rather than an RL learner. In this sense, it is suggested that the authors compare with T-LSA to justify the difference between RL and bandit (and perhaps non-myopic versus myopic), and with ALBL to justify the need of cross-dataset (which is readily done by the authors). Also, T-LSA needs some on-the-fly adaptive learning with bandit on the new data set while the proposed RL approach does not. So it would be interesting to know their differences.

---

> > > ### Author Response · Authors · 2018-01-23
> > > **Clarification.**
> > >
> > > Sorry about the confusion, this was our oversight. We will correct the inaccurate sentence.
> > >
> > > We also agree T-LSA is relevant for comparison, and we are running the experiment now and will add it to the final version. To contrast them explicitly, we expect MAP-GAL to perform better: (i) Due to non-myopic RL learning, (ii) Explicit dataset-adaptation mechanism multi-task trained on multiple sources (auxiliary network), rather than simply training a linear model using previous dataset parameters as regulariser. This also means queries are not wasted doing learning on the target problem. (iii) A deep policy rather than linear ensemble weighting.

---

### Author Response · Authors · 2018-01-05
**Summary of Changes in Revised Paper**

Dear Reviewers and Chairs,

Thanks to the reviewers for their comments.

The reviewers generally found our idea interesting, but had some questions and suggestions for improvements of the experiments.

We have revised our paper based on some of the reviewers' suggestions. Besides small clarifications and minor corrections, the main changes are as follows:
1. Based on R1 and R3's suggestion we have added two new baselines for comparison: ALBL & QBB.
2. R3 suggested cross-validation to avoid the bias of a fixed  train/test split and also asked about the dependence on the number of training datasets. To address this point thoroughly, we: (i) Re-organised the main experiment around leave-one-dataset out cross-validation, rather than a fixed 7/7 split. So now every dataset occurs as both a training and testing set. See updated Tab 1 & 2 in revised version. (ii) Explored the dependence on the number of training datasets. See Fig 2b in the revised version and the new Sec 7.2 for further discussion.
3. R1 wondered about the convergence process. This is now illustrated in Sec 7.1, Fig 3.

---

### Decision · Program_Chairs · 2018-01-29
**ICLR 2018 Conference Acceptance Decision**

**Decision:**

Reject

**Comment:**

In general, this seems like a sensible idea, but in my opinion the empirical results do not show a very compelling margin between using *entropy* as an active learning selection criterion vs the proposed methods. The difference is small enough that in practice it is very hard for me to believe that many researchers would choose to use the meta-learning via deep RL method (given that they'd need to train on multiple datasets and tune REINFORCE which is not going to be obviously easy). For that reason I am inclined to reject the paper.

In a follow-up version, I would heed the advice of Reviewer 1 and do more ablation analyses to understand the value of myopic vs non-myopic, cross-dataset vs. not, bandits vs RL, on the fly vs not (these are all intermingled issues). The relative lack of such analyses in the paper does not help in terms of it passing the bar.